# Potassium channel TASK-5 forms functional heterodimers with TASK-1 and TASK-3 to break its silence

Susanne Rinné [1,7], Florian Schick [1,7], Kirsty Vowinkel [1], Sven Schütte [1], Cornelius Krasel [2], Silke Kauferstein [3,4], Martin K.-H. Schäfer [5], Aytug K. Kiper [1], Thomas Müller [6] & Niels Decher [1] ✉

TASK-5 (*KCNK15*) belongs to the acid-sensitive subfamily of two-pore domain potassium ($K_{2P}$) channels, which includes TASK-1 and TASK-3. TASK-5 stands out as $K_{2P}$ channel for which there is no functional data available, since it was reported in 2001 as non-functional and thus "silent". Here we show that TASK-5 channels are indeed non-functional as homodimers, but are involved in the formation of functional channel complexes with TASK-1 and TASK-3. TASK-5 negatively modulates the surface expression of TASK channels, while the heteromeric TASK-5-containing channel complexes located at the plasma membrane are characterized by changes in single-channel conductance, Gq-coupled receptor-mediated channel inhibition, and sensitivity to TASK modulators. The unique pharmacology of TASK-1/TASK-5 heterodimers, affected by a common polymorphism in *KCNK15*, needs to be carefully considered in the future development of drugs targeting TASK channels. Our observations provide an access to study TASK-5 at the functional level, particularly in malignant cancers associated with *KCNK15*.

TASK-5 channels, encoded by *KCNK15*, are members of the acid-sensitive $K_{2P}$ channel subfamily of TASK channels, including TASK-1 and TASK-3. In humans, TASK-5 mRNA expression has been described in the adrenal gland, pancreas, liver, kidney, lung, ovary, testis, and heart[1,2]. TASK channels are increasingly recognized as key pharmacological targets for a wide range of human diseases, such as atrial fibrillation, sleep apnea, and pulmonary hypertension[3–9]. The TASK-5 channel is, on the other hand, one of the few $K_{2P}$ channels for which no functional data is available, as it was considered, by four independent studies describing the initial cloning[1,2,10,11], to be non-functional or an intracellular channel. As expected, a single nucleotide polymorphism in *KCNK15*, leading to the selectivity filter variant TASK-5[G95E], also failed to show channel activity[2]. Attempts to render TASK-5 channels

functional, for example by removal of a putative N-terminal endo-plasmic reticulum retention sequence[1], exchanging the C-terminus of TASK-3 and TASK-5 or recording at different extracellular pH values[2], failed to induce TASK-5 channel function. However, its distinct expression pattern in humans does not support the idea of TASK-5 being an intracellular ion channel, as those would rather show a homogeneous or widespread expression pattern.

Initial reports failed to indicate a heteromerization with TASK-1 or TASK-3, and it was concluded that TASK-5 does not form heterodimers within the TASK family[1,10]. Thus for over twenty years, it was hypothesized that TASK-5 may require a still unidentified accessory protein to form functional channels in the plasma membrane, or that it may form a channel in an intracellular organelle[1,2,10,11].

[1]Institute of Physiology and Pathophysiology, Vegetative Physiology, Philipps University Marburg, Marburg, Germany. [2]Institute for Pharmacology and Clinical Pharmacy, Faculty of Pharmacy, Philipps-University Marburg, Marburg, Germany. [3]Centre for Sudden Cardiac Death and Institute of Legal Medicine, University Hospital Frankfurt, Goethe-University, Frankfurt/Main, Germany. [4]DZHK (German Centre for Cardiovascular Research), Partner Site Rhein-Main, Frankfurt, Germany. [5]Institute of Anatomy and Cell Biology, Philipps University Marburg, Marburg, Germany. [6]Bayer AG, Research & Development, Pharmaceuticals, Wuppertal, Germany. [7]These authors contributed equally: Susanne Rinné, Florian Schick. ✉e-mail: decher@staff.uni-marburg.de

Here, we show that TASK-5 forms heteromeric channel complexes with TASK-1 and TASK-3. Co-expression of TASK-5 with TASK-1 or TASK-3 results in robust detection of TASK-5 containing heteromeric channels at the plasma membrane, whereas the number of homomeric TASK-1 and TASK-3 channels is reduced. Single-channel measurements reveal heteromeric TASK-3/TASK-5 channels with single-channel properties different from those of TASK-1 or TASK-3. Most importantly, TASK-5 channels dramatically alter TASK channel pharmacology and Gq-receptor-mediated inhibition of heteromeric TASK-1/TASK-5 channels. The unique pharmacology of TASK-1/TASK-5 heterodimers should be considered in the development of future drugs targeting TASK-1 channels in various cardiovascular diseases and different types of cancer.

## Results

### TASK-5 does not form homodimeric channels at the plasma membrane

Similar to previous reports[1,2,10,11], TASK-5 channels could not be recorded in *Xenopus* oocytes (Fig. 1a, b). Therefore, we employed in the current study a more comprehensive approach to functionally record TASK-5 channels, including the testing of a number of specific mutants (Fig. 1c). First, we examined whether a more alkaline extracellular pH than previously studied[1,2] might result in channel function. However, even very alkaline solutions, with a pH of up to 10.5, did not lead to functional expression (Fig. 1d).

We recently described an inner gate in TASK-1 channels, the so-called "X-gate" structure, which stabilizes the closed state of the channel[12]. As the amino acid sequence of the X-gate region is highly similar within the TASK family, it appears very likely that TASK-5 may also contain an X-gate-like structure, and consequently, TASK-5 might be non-functional as it is primarily in a very stable closed state. Therefore, we studied analogous mutations in the putative X-gate of TASK-5 (TASK-5[R7D] or TASK-5[R131D], Fig. 1c and Supplementary Figs. 1 and 2), which resulted in a strong destabilization of the inner gate and a gain-of-function in TASK-1. However, the corresponding mutations did not lead to functional TASK-5 expression (Fig. 1e).

Given that a cysteine residue at the tip of the cap structure was discussed as relevant for the dimerization of some $K_{2P}$ channels[13,14], we introduced a cysteine residue at the corresponding amino acid 53 in TASK-5. However, this TASK-5[G53C] mutant also failed to produce significant currents (Fig. 1f). Moreover, it has been proposed that several amino acid residues at the tip of the cap structure of TASK-1 ([52]YNLS[55]) are essential for functional expression[14]. To convert the tip of TASK-5 into a more TASK-1-like cap structure and compensate for putative limitations in the self-assembly or stability of the channel, we introduced a [52]FGFS[55] to [52]YNLS[55] exchange in the TASK-5 cap structure (Fig. 1c and Supplementary Fig. 1). However, this TASK-5 channel construct remained also non-functional (Fig. 1g).

Most of the approaches to functionally express TASK-5 were performed with oocyte storage solutions that contained the antibiotic gentamycin and/or the phosphodiesterase inhibitor theophylline. However, also removing those drugs from the storage solution did not induce any TASK-5-mediated currents (Fig. 1h).

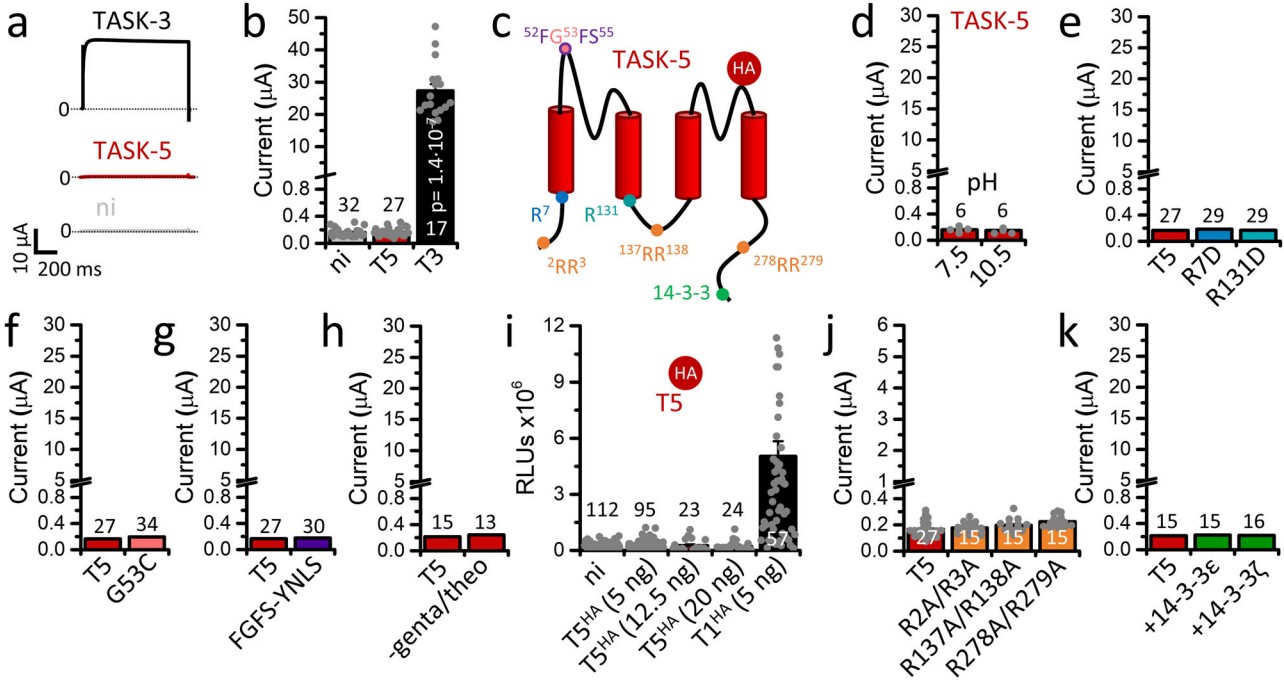

**Fig. 1 | TASK-5 resists to form homomeric channels at the plasma membrane.**
**a** Representative current traces of TASK-3 (black) and TASK-5 (red) recorded in *Xenopus laevis* oocytes after injection of 25 ng TASK-5 or 50 pg TASK-3. A voltage-step protocol from a holding potential of −80 mV to +40 mV was applied. ni: non-injected oocytes. **b** Mean current amplitudes analyzed at +40 mV. Significance was probed by comparing TASK-3 (black) and TASK-5 (red). **c** Cartoon of a TASK-5 subunit, illustrating the location of amino acid mutations, potential trafficking signals, and the localization of the extracellular hemagglutinin (HA)-epitope (red) that was introduced in individual colors. Transmembrane domains are highlighted in red. **d** Mean current amplitudes of TASK-5 injected oocytes at extracellular pH 7.5 or 10.5. **e** Mean current amplitudes after injection of 25 ng TASK-5 (red) or the X-gate mutants TASK-5[R7D] (blue) or TASK-5[R131D] (turquoise). **f** Mean current amplitudes of cap structure mutants TASK-5[G53C] (light red) or **g** TASK-5[FGFS-YNLS] (purple).

**h** Mean current amplitudes of TASK-5, recorded after storage of injected oocytes in solution with or without gentamycin and theophylline. **i** Analysis of the surface expression of TASK-5 (red), after injection of 5, 12.5, or 20 ng TASK-5[HA-Ex] cRNA per oocyte. The HA symbol indicates, that TASK-5 was extracellularly HA-tagged. RLUs: relative light units. TASK-1[HA-Ex] (black, 5 ng/oocyte) served as a positive control. **j** Mean current amplitudes of TASK-5 (red) with mutations of cytosolic retention signals (light orange): TASK-5[R2A/R3A] (N-terminus), TASK-5[R137A/R138A] (M2-M3 linker) or TASK-5[R278A/R279A] (C-terminus). 25 ng cRNA of each mutant was injected per oocyte. **k** Mean current amplitudes after co-expression of TASK-5 (red) with 14-3-3ε or 14-3-3ζ (green). Number of experiments (biological replicates) are provided within the bar graphs. Data are presented as mean ± s.e.m. Significance was probed using the Mood's median test (two-sided). Data were considered as significant with a confidence interval of 95% ($p < 0.05$). Source data are provided as a Source Data file.

Studying the surface expression of TASK-5, which contains an extracellular HA-epitope (TASK-5[HA-Ex]) (Fig. 1c and Supplementary Fig. 1) in *Xenopus laevis* oocytes using an ELISA-based luminometric assay, the channel was not detected at the plasma membrane. In contrast, TASK-1[HA-Ex], which served as a positive control[15], was robustly expressed on the cell surface (Fig. 1i). In fluorescence imaging experiments, TASK-5 was localized in intracellular compartments and exhibited a typical web-like fluorescence pattern, which suggests that it is primarily located in the ER (Supplementary Fig. 3). The idea that TASK-5 channels get stuck in the ER, was previously probed by mutating the putative N-terminal ER retention signal $^2$RR$^{31}$. Here, we reinvestigated the role of $^2$RR$^3$ and additional putative di-arginine signals that were predicted in the M2-M3 linker at position $^{137}$RR$^{138}$, as well as in the cytosolic C-terminus at position $^{278}$RR$^{279}$ (Fig. 1c and Supplementary Fig. 1). However, mutating these putative ER retention signals did not result in the functional expression of TASK-5 (Fig. 1j). As TASK-5 also contains a putative C-terminal motif for 14-3-3 binding (Fig. 1c and Supplementary Fig. 1), we co-expressed 14-3-3ε or 14-3-3ζ with TASK-5 in order to measure TASK-5 channels at the plasma membrane. However, even after co-expression with 14-3-3, we were not able to measure any TASK-5 currents (Fig. 1k).

## TASK-5 forms functional heterodimers with TASK-1 and TASK-3, changing channel composition at the plasma membrane

TASK-5 is co-expressed with TASK-1 and TASK-3 in the human heart[1,2,4], pulmonary vascular smooth muscle cells (VSMCs)[16], respiratory neurons[10], as well as the nasopharynx (human protein atlas)[17]. These co-expression patterns are highly relevant for the development of TASK-1 modulators against atrial fibrillation (AFib)[3–5,18], obstructive and central sleep apnea (OSA and CSA)[3,7] or pulmonary arterial hypertension (PAH)[8,19–21]. This prompted us to re-investigate whether TASK-5 channels heteromerize with other TASK channel family members, although heteromerization was previously not observed[1,10].

Heteromerization within the family of acid-sensitive $K_{2P}$ channels, namely between TASK-1 and TASK-3, has already been described[22–24]. Here, we tested whether TASK-5 might after all form functional heterodimers with its subfamily members TASK-1 and/or TASK-3. First, we co-expressed TASK-1 (Fig. 2a, b) or TASK-3 (Fig. 2c, d) with increasing amounts of TASK-5. In both cases, TASK-5 expression decreased current amplitudes in a concentration-dependent manner (Fig. 2a–d). Next, we analyzed the effects of TASK-5 on the surface expression of TASK-1[HA-Ex] or TASK-3[HA-Ex], using constructs that are non-functional, but very well suited for studying the surface expression of the channels[15]. Here, we found that the current reduction corresponded to a reduced surface expression of both channels in the presence of TASK-5 (Fig. 2e, f). As a control, co-expression of TASK-3 with TASK-1[HA-Ex] led to an increased surface expression of TASK-1 (Fig. 2e), while co-expression of TASK-1 with TASK-3[HA-Ex] did not alter the robust TASK-3 signal at the plasma membrane (Fig. 2f), as previously described[24].

Next, we examined whether TASK-1 or TASK-3 might assist TASK-5 in reaching the cell surface. To this end, we co-expressed an extracellularly HA-tagged TASK-5 channel construct (TASK-5[HA-Ex]) with non-tagged TASK-1 (Fig. 2g) or TASK-3 (Fig. 2h). Co-expression with TASK-1, as well as TASK-3, resulted in a pronounced localization of TASK-5[HA-Ex] at the plasma membrane (Fig. 2g, h), which was further augmented in a concentration-dependent manner for both channels (Fig. 2g, h). In summary, the reduction in TASK currents induced by TASK-5 correlated with reduced surface expression, while conversely TASK-5 surface expression was increased (Supplementary Fig. 4).

Co-expression with selectivity filter defect mutants, in which the highly conserved GYG motif within the pore signature sequence of potassium channels is changed to EYG, is a commonly used approach to probe for heteromerization of $K_{2P}$ channels[25–27]. As heteromerization of TASK-1 and TASK-3 is convincingly described, we first performed

control-experiments co-expressing wild-type TASK-3 channels or the dominant-negative TASK-3[G95E] mutant with TASK-1 (Fig. 2i). As previously described[24,25], TASK-3 co-expression with TASK-1 led to a drastic increase in channel amplitude compared to TASK-1 injected alone (Fig. 2i). In addition, co-expression of the TASK-3[G95E] mutant with TASK-1 led to the emergence of strong dominant-negative effects, with a concentration-dependent reduction in current (Fig. 2i). Similarly, TASK-3[G95E] lead to drastic decreased current amplitudes of TASK-3 (Fig. 2j). These findings corroborate previous data and validate our experimental approach. In the same experimental setting, TASK-5[G95E] did not conduct currents on its own (Fig. 2k, l), while it reduced the current amplitudes of TASK-1 and TASK-3 in a dominant-negative manner. These effects were significantly stronger than those of wild-type TASK-5 (Fig. 2k, l). In summary, these findings further support the hypothesis that TASK-5 forms functional heterodimers with TASK-1 and TASK-3, changing channel composition at the cell surface.

## Single-channel patch-clamp recordings support the hypothesis of the formation of heteromeric TASK-5 channel complexes

To further substantiate our initial findings that TASK-5 forms functional heterodimers with other TASK channel family members, we conducted inside-out single-channel patch-clamp recordings in *Xenopus* oocytes in search of single-channel events characterized by heteromeric channel complexes containing TASK-5. Given that TASK-3 has a larger single-channel conductance and longer open times than TASK-1, it is more feasible to perform such a single-channel analysis with TASK-3. Here, TASK-3 homodimers yielded characteristic single-channel events with an average amplitude of 8.2 pA at −100 mV (Fig. 3a, b), whereas co-expression of TASK-3 with TASK-5 led to the emergence of a novel additional conductance with a single-channel amplitude of 5.5 pA at −100 mV (Fig. 3a, b), presumably reflecting the formation of TASK-3/TASK-5 heterodimers. Figure 3a illustrates representative recordings from patches containing either (i) TASK-3, (ii) TASK-3 together with heteromeric TASK-3/TASK-5 channels, or (iii) only heterodimeric TASK-3/TASK-5 channels. For the recordings categorized as TASK-3 together with heteromeric TASK-3/TASK-5 channels, we can exclude that they contained only one heteromeric TASK-3/TASK-5 channel with a subconductance state, because we sometimes observed cumulative opening events of two channels (Supplementary Fig. 5). Following the injection of TASK-3 cRNA alone, a TASK-3-like channel was observed in 15 out of 15 patches (Fig. 3c). The co-expression of TASK-3 and TASK-5 yielded TASK-3-like events in 18 out of 27 patches, while in 9 out of 27 patches a heterodimeric TASK-3/TASK-5-like channel was observed (Fig. 3c). In 5 of these 9 patches, solely the heteromeric channel with a single-channel conductance (SCC) of 93 pS was detected (Fig. 3c, d), whereas the SCC of TASK-3 was 113 pS (Fig. 3d). While the open time $\tau_o$ of the heteromer-like channels ($\tau_o = 1.98$ ms) was highly similar to that of TASK-3 channels ($\tau_o = 1.92$ ms) (Fig. 3e), there were major differences in the closed times (Fig. 3f, g). Here, the short closed time ($\tau_{C1}$) was prolonged from 55.1 ms for TASK-3 to 171.3 ms for the heteromeric channels and the long closed time ($\tau_{C2}$) from 340.6 ms to 982.0 ms, respectively (Fig. 3f,g). Consistent with this prolonged closed time, the $NP_o$ of heteromeric TASK-3/TASK-5-like channels was very low (≈0.01%) and about one-third of that of TASK-3.

In detail, the reduction of the single-channel conductance, the increased closed times together with the reduced surface expression, contribute to the reduced current amplitudes of TASK-5 heterodimers with TASK channels recorded in voltage-clamp experiments. Taken together, the patch-clamp experiments, revealing an additional conductance with specific single-channel properties, strongly support the assumption of the formation of functional TASK-3/TASK-5 channel complexes that show characteristic single-channel properties.

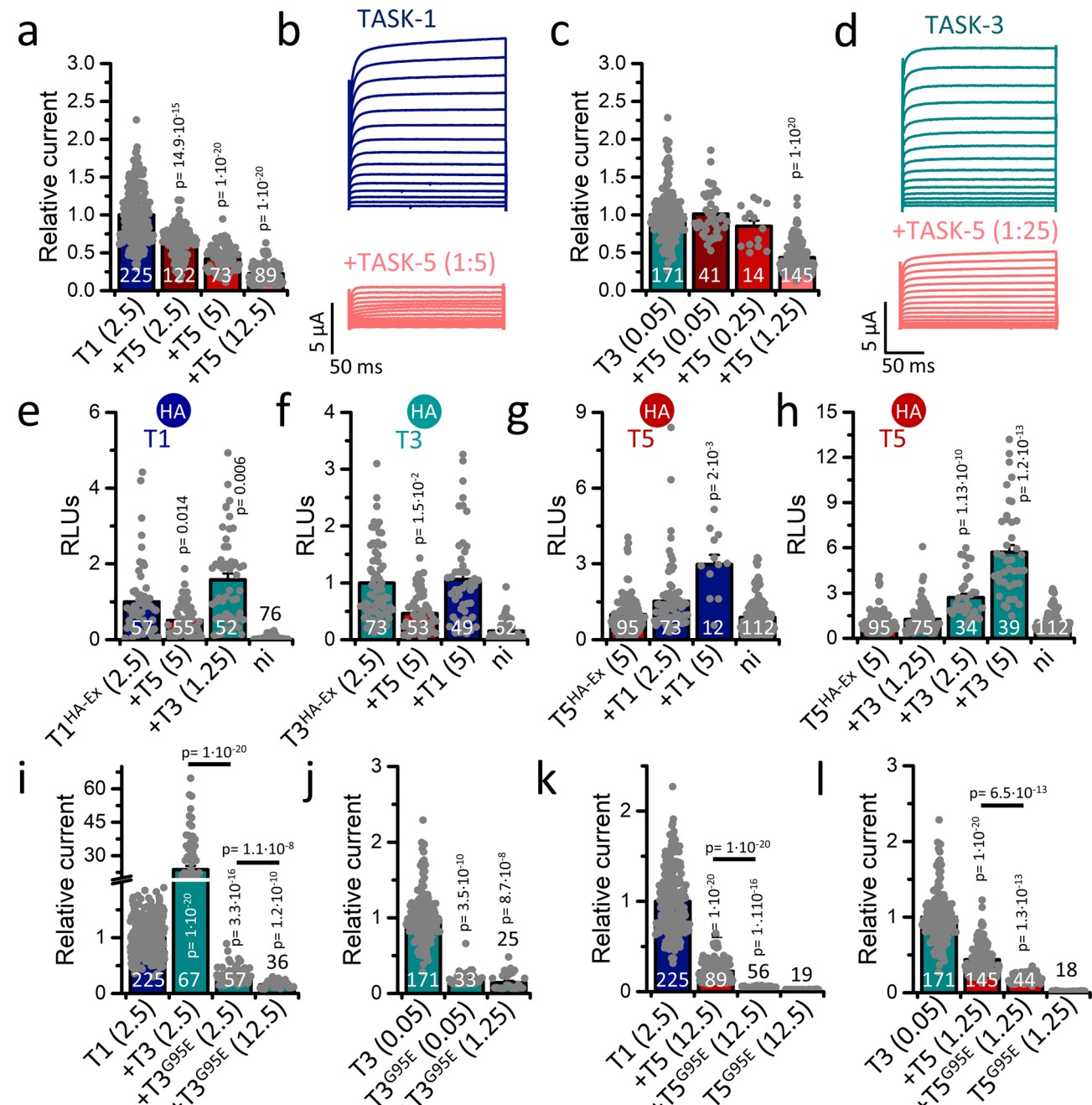

**Fig. 2 | TASK-5 forms functional heterodimers with TASK-1 and TASK-3.**
**a** TASK-1 (blue) or **c** TASK-3 (green) expressed alone or co-expressed with TASK-5 (red) in *Xenopus* oocytes. Currents were analyzed at +40 mV and normalized to TASK-1 or TASK-3, respectively. Significance was probed compared to TASK-1 or TASK-3. **b** Representative current traces of TASK-1 or **d** TASK-3 alone or co-expressed with TASK-5. Currents were recorded by stepping from −80 mV to potentials ranging from −70 to +70 mV. **e** Quantification of TASK-1[HA-Ex] or **f** TASK-3[HA-Ex] surface expression in an ELISA-based chemiluminescence assay alone or together with either TASK-5 or **e** TASK-3 and **f** TASK-1. **g** Analysis of TASK-5[HA-Ex] surface expression after injection of TASK-5[HA-Ex] alone or together with TASK-1 or **h** TASK-3, respectively. The respective extracellularly hemagglutinin (HA)-tagged channel construct used is indicated by the HA symbol. Significance was probed

compared to the respective HA-tagged channel expressed alone. **i** Relative current amplitudes of TASK-1 or **j** TASK-3 expressed alone or co-expressed with TASK-3 or TASK-3[G95E]. If not indicated, significance was probed compared to TASK-1 or TASK-3 expressed alone. **k** Relative current amplitudes of TASK-1 or **l** TASK-3 expressed alone or co-expressed with TASK-5 or TASK-5[G95E]. cRNA amount injected is shown in ng/oocyte in brackets. Numbers of experiments (biological replicates) are displayed within the bars. Data are presented as mean ± s.e.m. Significance was probed using Mood's median test (two-sided), *p*-values are given within the graphs. Significance was probed compared to the respective channel expressed alone. Data were considered as significant with a confidence interval of 95% (*p* < 0.05). Source data are provided as a Source Data file.

## TASK-5 alters the gating of TASK heterodimers

The results described above reveal an altered gating of the heteromeric channel complexes containing TASK-5. Thus, we proceeded to investigate whether also TASK-5-dependent changes in the extracellular pH gating were evident in heteromeric complexes with TASK-1

and TASK-3 (Fig. 4a, f). However, co-expression of TASK-5 did not result in significant alterations in the pH50 values of TASK-1 (Fig. 4a) or TASK-3 (Fig. 4f). Histidine 98 is the extracellular pH sensor in TASK-1[28] and TASK-3[29,30]. Since the His 98 is also conserved in TASK-5, it seems to be plausible that TASK-5 does not influence the pH sensitivity of

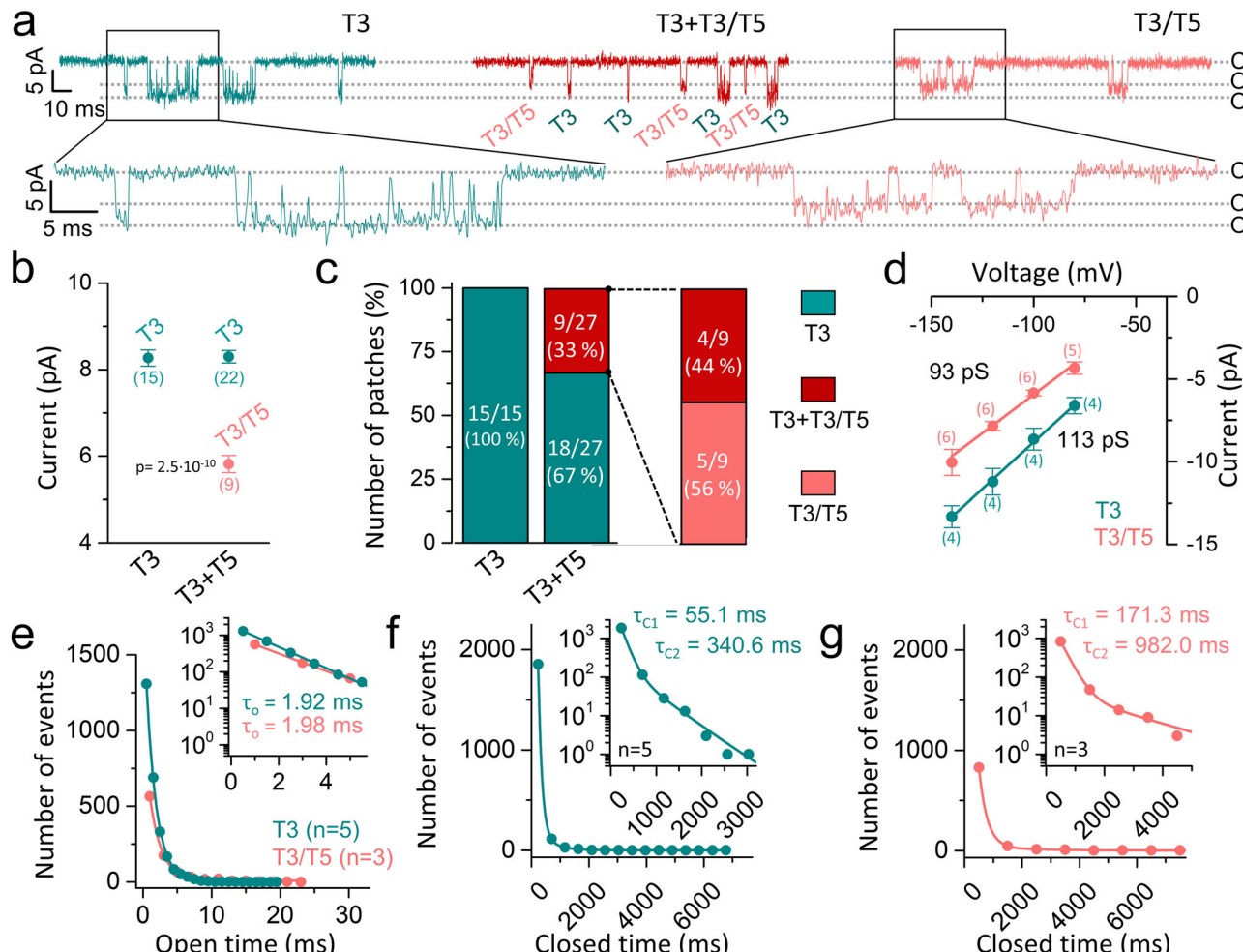

**Fig. 3 | Single-channel patch-clamp recordings of heteromeric TASK-5 channel complexes. a** Patches with currents representing TASK-3 (T3, left) alone, TASK-3 and heteromeric TASK-3/5 (T3 + T3/T5, middle) or exclusive TASK-3/5 currents (T3/T5, right), recorded at −100 mV. **b** Single-channel amplitudes of TASK-3 or TASK-3/TASK-5 heterodimers. The numbers of patches are provided underneath the respective data points. Significance was probed using a two-sided unpaired Student's *t*-test comparing TASK-3/TASK-5 with TASK-3. **c** Number of patches containing TASK-3 alone (T3, green), TASK-3 plus TASK-3/TASK-5 heterodimers (T3 + T3/T5, red) or TASK-3/TASK-5 heterodimers alone (T3/T5, light red), after injection of TASK-3 cRNA (left bar) or co-injection of TASK-3 with TASK-5 (middle bar). Right bar illustrates the channel distribution in heterodimer-containing patches. **d** Analysis of the single-channel conductances (number of patches is indicated in brackets) and **e** open times $\tau_o$ of TASK-3 homodimers and TASK-3/5 heterodimers (number of patches as in **d**). **f** Analysis of short ($\tau_{C1}$) and long closed times ($\tau_{C2}$) of TASK-3 or **g** TASK-3/5 ($n = 3-5$). Data are presented as mean ± s.e.m. *p*-values are given within the graphs. Data were considered as significant with a confidence interval of 95% ($p < 0.05$). Source data are provided as a Source Data file.

TASK-1 or TASK-3 when it is part of the heteromeric channel complex. As a role in thermosensation was proposed for TASK-1 and TASK-3[31,32], we also analyzed the temperature-dependent gating of TASK-1 or TASK-3 heterodimers containing TASK-5. However, TASK-5 did not influence the thermosensitivity ($Q_{10}$ values) of TASK channel complexes (Supplementary Fig. 6).

The single-channel recordings of TASK-3/TASK-5 heteromeric channels revealed prolonged closed times, indicating a stabilization of the closed state, which we postulated recently for TASK-1 to occur at the so-called X-gate[12]. Residues R7 and R131 are essential for the stabilization of the inner gate of TASK-1. Mutation of these residues led to a strong increase in TASK-1 currents[12], but not in homodimeric TASK-5 channels (Fig. 1e).

Therefore, we sought to determine whether these non-functional TASK-5 mutants exert effects on the putative X-gate in heteromeric channel complexes with TASK-1 or TASK-3. To this end, we co-expressed TASK-5[R7D] or TASK-5[R131D] with TASK-1 (Fig. 4b–d) or TASK-3 (Fig. 4g–i). In the case of TASK-1, both mutants led to a significant increase in current amplitudes compared to the co-expression of wild-type TASK-5 (Fig. 4b, c), while the surface expression was not significantly altered (Fig. 4d). These data further add evidence that TASK-5 is involved in heteromeric complexes with TASK-1 and that these channels may have an inner X-gate-like structure that can be destabilized by mutants, similar as in TASK-1 channels. TASK-5[R7D] also resulted in a major increase in current amplitudes for heteromeric channels with TASK-3, whereas TASK-5[R131D] did not alter the gating (Fig. 4g–i). Thus, TASK-5 heteromerization with TASK-1 and TASK-3 was again evident, however, the putative X-gate-like structure of heteromeric TASK-5 channels is presumed to diverge between TASK-1 and TASK-3.

Given that Gq-coupled inhibition of homomeric TASK-1 and TASK-3 channels mechanistically occurs at this inner gate[33,34], we next examined the Gq-coupling of TASK-5-containing heterodimers. To compare the receptor-mediated channel inhibition via the application of the $\alpha_1$-adrenoreceptor agonist methoxamine (10 µM), we co-expressed TASK-1 or TASK-3 together with TASK-5 and the $\alpha_1$-adrenoreceptor (Fig. 4e, j). As previously described[35], TASK-1 current amplitudes were strongly diminished due to Gq-coupled receptor

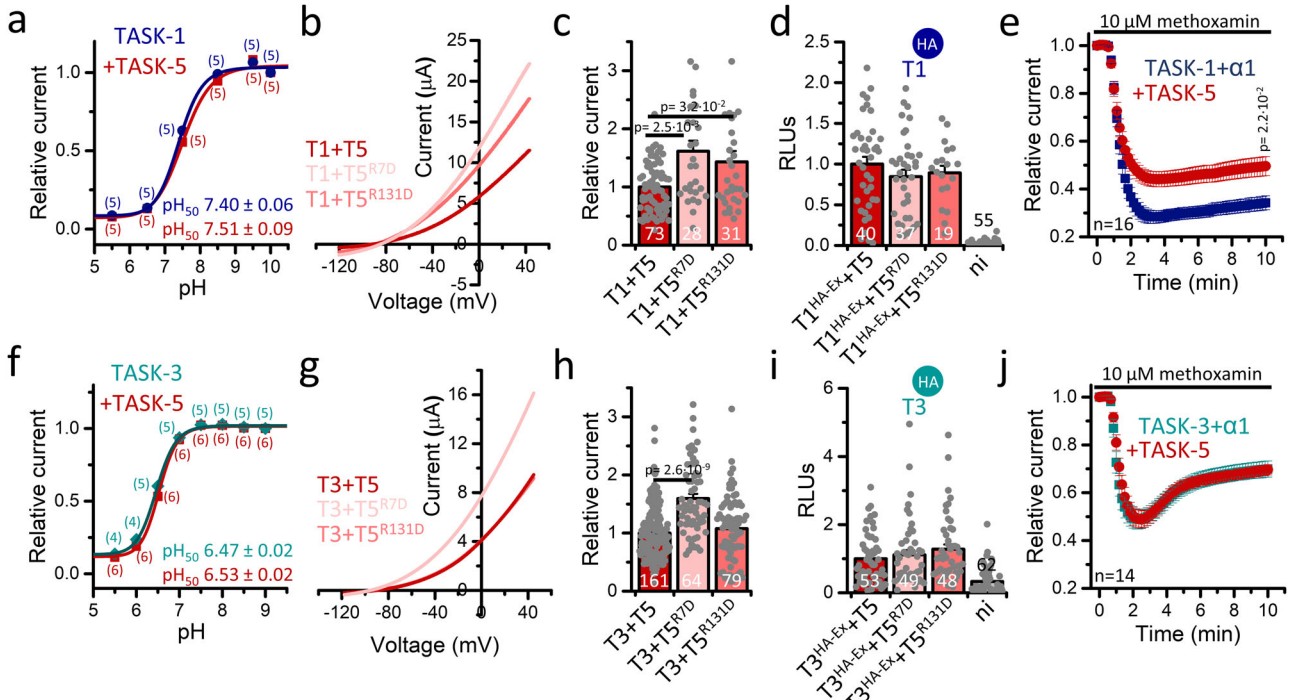

**Fig. 4 | TASK-5 alters the gating of TASK heterodimers. a** Effects of extracellular pH changes on relative currents of TASK-1 (blue) and TASK-1/5 (red) or **f** TASK-3 (green) and TASK-3/5 (red) expressing oocytes. Numbers of measurements of independently injected oocytes (biological replicates) are shown in brackets. Data were analyzed at +40 mV and normalized to currents at pH 10. **b** Representative current traces of either TASK-1 or **g** TASK-3, respectively co-expressed with wild-type TASK-5 (red) or putative TASK-5 X-gate mutants (TASK-5[R7D] or TASK-5[R131D], light red). **c, h** Current amplitudes of (**b** and **g**, respectively) analyzed at +40 mV and normalized to wild-type current amplitudes. Numbers of biological replicates are given in the bar graphs. **d, i** Surface expression of TASK[HA-Ex] after co-expression with either wild-type TASK-5 or the putative X-gate mutants TASK-5[R7D] or TASK-5[R131D]. The respective extracellularly hemagglutinin (HA)-tagged channel construct used is

indicated by the HA symbol. Numbers of biological replicates are given in the bar graphs. **e** Gq-coupled receptor-mediated inhibition of TASK-1 or **j** TASK-3 and respective TASK-5 heteromeric channel complexes via co-expressed adrenergic $\alpha_1$ receptors. Currents were recorded by stepping from −80 to +40 mV. Numbers (n) of measurements of independently injected oocytes (biological replicates) are illustrated. Data are presented as mean ± s.e.m. Significance was probed using the two-sided Welch's *t*-test comparing T1 + T5 with the respective mutant (panel c) or two-sided Mood's median test (**e** comparing TASK-1 co-expressed with the $\alpha_1$ receptor with the co-expression of TASK-5 and **h** comparing T3+T5 with the respective TASK-5 mutant), *p*-values are given within the graphs. Data were considered as significant with a confidence interval of 95% ($p < 0.05$). Source data are provided as a Source Data file.

activation (Fig. 4e), while this reduction was less efficient and somewhat transient for TASK-3 channels (Fig. 4j). It is noteworthy that co-expression of TASK-5 did not alter the Gq-receptor-mediated inhibition of TASK-3 channels (Fig. 4j), whereas the receptor-mediated inhibition of heteromeric TASK-1/TASK-5 channels was strongly reduced (Fig. 4j). Similarly to the different effects of the TASK-5 X-gate mutants on TASK-1 and TASK-3, heteromeric TASK-1 or TASK-3 channel complexes containing TASK-5 subunits exhibited differential behavior concerning Gq-coupling, indicating a structural variability at the inner gate, which may be responsible for these variances we have observed here. The reduced Gq-coupling, which is specific to heteromeric TASK-1/TASK-5 channel complexes, might be of physiological relevance in tissues where these channels are co-expressed.

## TASK-5/TASK-1 heterodimers exhibit specific blocker affinities

TASK-1 blockers are promising drugs for the treatment of AFib[3–5,18] and OSA/CSA[3,7] and several TASK-1 blockers are undergoing clinical trials for the treatment of OSA (BAY2586116) and AFib (A293 = AVE1231, doxapram). Moreover, TASK-1 activators are predicted to be highly beneficial for the treatment of both heritable and non-heritable forms of PAH[8,19–21]. Thus, it is of the highest interest, whether the pharmacology of TASK-1/TASK-5 heterodimers is different compared to that of TASK-1 homodimers. Therefore, we tested a range of TASK-1 blockers (Fig. 5a–d, Supplementary Figs. 7 and 8). For BAY1000493, BAY2586116, and A293, an unexpected and highly pronounced increase in sensitivity (up to nearly 30-fold) was observed for the

TASK-1/TASK-5 heteromeric channels (Fig. 5a, b). In contrast, A1899 sensitivity was not different compared to TASK-1 (18.4 nM versus 27.4 nM) (Fig. 5a, b). These data suggest that TASK-1/TASK-5 heterodimers have a pharmacology that strongly differs from that of TASK-1, with potentially significant implications for current and future clinical studies. Highly potent TASK-1 blockers are trapped in the central cavity[12], whereas A1899, which additionally binds to residues below the X-gate, is not[36,37]. Thus, the enhanced affinity for selected TASK-1 blockers might be attributed to more pronounced trapping in heteromeric channels, which is consistent with the altered gating at the X-gate (increased closed times, reduced Gq-coupling) of heteromeric channels.

Systematic studies of drug-binding sites in TASK-1 revealed that L122, located in the M2 segment and facing the central cavity underneath the pore (Supplementary Fig. 2), is an essential residue for the binding of highly potent TASK-1 blockers[12,37]. This leucine residue is highly conserved within the TASK subfamily. Therefore, we sought to determine whether the TASK-5[L122A] mutant, which does not conduct currents in homomeric channels, might influence the drug affinity in heteromeric channels with TASK-1. For BAY1000493 and A293, a markedly diminished drug sensitivity was observed, accompanied by a 21- and 28-fold shift in the IC50, respectively (Fig. 5c, d). In contrast, for BAY2586116 and A293, almost no changes in drug sensitivity were observed (Fig. 5c, d). The fact that the co-expression of a silent channel, which harbors a mutation that faces the central cavity, alters the pharmacology of drugs that bind to the central cavity of TASK-1,

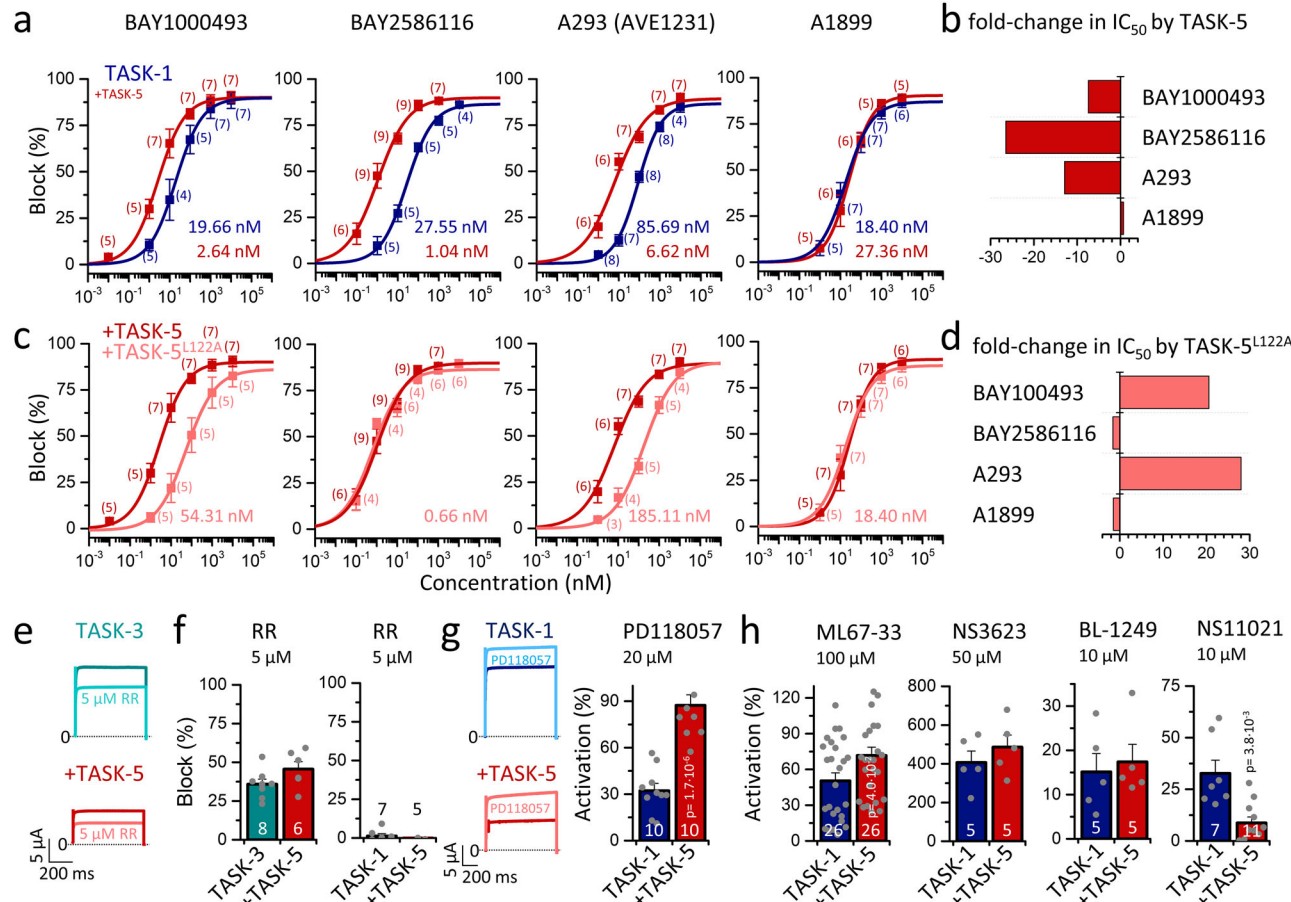

**Fig. 5 | TASK-1/5 heterodimers show specific pharmacological properties.**
**a** Dose-response curves of BAY1000493, BAY2586116, A293, and A1899 on TASK-1 homodimers (blue) or after co-expression with TASK-5 (red) and **c** in comparison with the putative pore facing TASK-5$^{L122A}$ mutant (light red). The IC$_{50}$ was determined from Hill plots. Numbers of biological replicates are given in brackets. **b** Fold-change in IC$_{50}$ of respective compounds for TASK-1/5 versus TASK-1. **d** Fold-change in IC$_{50}$ of heteromers containing the TASK-5$^{L122A}$ mutant, compared to wild-type TASK-5 heterodimers. **e** Block of TASK-3 (green) or heteromeric TASK-3/5 channels (red) by 5 μM ruthenium red (RR), recorded by applying voltage-steps to +40 mV. **f** Block of TASK-3 (green) and TASK-3/5 (red, left) or TASK-1 (blue) and TASK-1/5 (red, right) currents by 5 μM RR. **g** Activation of TASK-1 (blue) or heteromeric TASK-1/5 channels (red) by 20 μM PD118057, recorded by applying voltage-steps to +40 mV. Activation of TASK-1 (blue) or TASK-1/5 currents (red) by 20 μM PD118057. Significance was probed by comparing TASK-1 with TASK-1/5. **h** Activation of TASK-1 and TASK-1/5 currents by ML67-33, NS3623, BL-1249, or NS11021. Significance was probed by comparing TASK-1 with TASK-1/5. Data are presented as mean ± s.e.m. Significance was probed using two-sided unpaired Student's $t$-test (**g**) or two-sided Mann–Whitney U-test (**h**), $p$-values are given within the graphs. Data were considered as significant with a confidence interval of 95% ($p < 0.05$). Source data are provided as a Source Data file.

further corroborates the formation of functional TASK-5 containing heteromeric channel complexes.

Ruthenium Red (RR) is a valuable tool that is commonly used to discriminate currents in native tissue that are mediated by TASK-3 from those conducted by TASK-1 since only TASK-3 is RR sensitive[38]. Czirják et al. reported that the cationic dye inhibited homomeric TASK-3 channels, whereas TASK-1 homodimers and TASK-1/TASK-3 heterodimers were not affected, as two E70 residues are necessary (one in each subunit) for the inhibition by this compound[38] and only TASK-3 contains a glutamate at this position (K70 in TASK-1). TASK-5 has an 'E' at this Keystone inhibitor site (just like E70 in TASK-3) which is known to be important for RR binding[39]. Thus, a TASK-3/TASK-5 heterodimer is expected to have an RR-competent site (E/E). The heterodimer mixture (K/E) in TASK-1/TASK-3 (not RR sensitive[38]) and in TASK-1/TASK-5 heterodimers, is expected to have an RR-incompetent site. Consistently, TASK-3/TASK-5 heterodimers and homomeric TASK-3 exhibited equivalent sensitivity to RR (5 μM) (Fig. 5e,f), while TASK-1 and TASK-1/TASK-5 heterodimers demonstrated no RR-sensitivity (Fig. 5e,f). Therefore, RR cannot be utilized to identify heteromeric TASK channel complexes with TASK-5 in native tissue, since TASK-5 subunits do not introduce RR-sensitivity into TASK-1/TASK-5

channel complexes or remove RR-sensitivity of heteromeric TASK-3/ TASK-5 complexes (Fig. 5e, f).

## TASK-5/TASK-1 heterodimers exhibit a specific affinity for channel activators

K$_{2P}$ channels can be activated at the level of the selectivity filter by so-called negatively charged activators (NCAs)[40]. However, it was not reported whether TASK-1 channels respond to NCAs, despite these channels possess an additional inner gate. TASK-1 activating compounds might be, however, promising novel drugs for the treatment of heritable and non-heritable forms of PAH[8,19–21]. Thus, we tested different NCAs and found that they also activate TASK-1 (Fig. 5g, h), albeit less efficiently than other K$_{2P}$ channels[40].

Perfusion with PD118057 resulted in potent activation of heteromeric TASK-1/TASK-5 channel complexes, while there was only a minor activation of homomeric TASK-1 channels (Fig. 5g). Also, ML67-33, originally discovered as a potent activator of TREK-1 channels[41], caused a more pronounced activation of heteromeric TASK-1/TASK-5 channels (Fig. 5h), whereas for other NCAs (NS3623 or BL-1249) a comparable efficacy in activating heteromeric channels containing TASK-5 or less activation (NS11021) (Fig. 5h) was noted. In conclusion, TASK-5

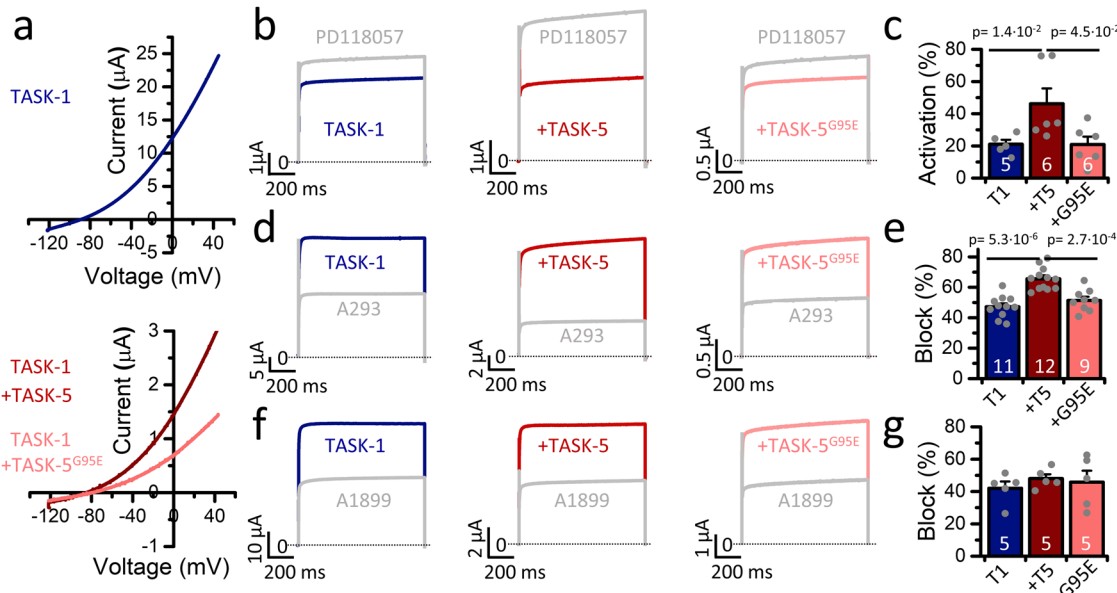

**Fig. 6 | TASK-5G95E polymorphism alters drug sensitivity of heteromeric TASK-1/5 channel complexes. a** Representative current traces of TASK-1 homodimers (blue, upper panel) and TASK-1 co-expressed with either TASK-5 or TASK-5$^{G95E}$ (red and light red, lower panel). Currents were recorded by a voltage ramp from −120 to +45 mV. **b** Respective current traces under control conditions (blue) and application of 20 μM PD118057, **d** 100 nM A293 or **f** 40 nM A1899 (gray) and

**c**, **e**, **g** corresponding percentage of activation/block. Data are presented as mean ± s.e.m. Significance was probed using the two-sided Mann–Whitney U-test (**c**) or two-sided unpaired Student's t-test (**e**). Significance was probed as indicated by the bars. Data were considered as significant with a confidence interval of 95% (p < 0.05). Source data are provided as a Source Data file.

containing heteromers show a unique pharmacology for blockers and activators that should be carefully considered in future studies on basic research, as well as in the research and development of novel drugs targeting TASK channels.

### The TASK-5$^{G95E}$ polymorphism alters drug sensitivity of TASK-1 toward channel blockers

About half of the general population carries a TASK-5$^{G95E}$ polymorphism in one allele (46%), while about 31% are homozygous for this variant (GnomAD Exac database). This polymorphism corresponds to the non-conducting TASK-5$^{G95E}$ mutant with a non-functional selectivity filter (GYG sequence is changed to EYG)[10]. As described above, this variant acts in a dominant-negative manner. The co-expression of TASK-1 with TASK-5$^{G95E}$, which reflects the homozygous state, resulted in a more pronounced current suppression than by wild-type TASK-5 (Fig. 2k). It is therefore pertinent to inquire whether homozygous polymorphism carriers may exhibit altered drug-sensitivities, as the non-conducting TASK-5$^{G95E}$ subunits may be unable to confer higher drug-sensitivities to TASK-1, as was demonstrated for wild-type TASK-5 subunits (Fig. 5). To address this question, we expressed TASK-1 (Fig. 6a, upper panel), TASK-1 with TASK-5 (Fig. 6a, lower panel) or TASK-1 with TASK-5$^{G95E}$ (Fig. 6a, lower panel) in oocytes and applied the activator PD118057 (Fig. 6b). While currents after co-expression of TASK-1/TASK-5 were activated by about 45%, currents after co-expression of TASK-1/TASK-5$^{G95E}$ were only activated by 21% (Fig. 6b, c). Similarly, the TASK blocker A293 demonstrated significantly reduced effects on TASK-5$^{G95E}$ containing heterodimers in comparison to wild-type heteromeric TASK-1/TASK-5 channels (Fig. 6d, e). As a control, co-expression of TASK-1 with TASK-5$^{G95E}$ did not alter drug affinity to A1899 (Fig. 6f, g), a drug that did not show an increased affinity for heteromeric TASK-1/TASK-5 channels (Fig. 5a, b). In conclusion, the enhanced drug sensitivity of TASK-5-containing heteromeric channels may be subject to alteration for some drugs in native tissue, particularly in the case of homozygous polymorphism carriers, which needs to be considered in future clinical trials and/or pharmacogenomics studies.

### Discussion

Since its cloning two decades ago, TASK-5 has been presumed to be non-functional and dimerization with other TASK channels was excluded[1,10]. Therefore, the heteromerization of silent TASK-5 channels with TASK-1 or TASK-3 was not further considered. Our study provides compelling evidence that TASK-5 reaches the plasma membrane when co-expressed with TASK-1 or TASK-3 channels and is indeed part of functional heteromeric TASK channel complexes at the plasma membrane. Consequently, all attempts in the past, to functionally express homomeric TASK-5 channels had to fail, if the channel was only active in heteromeric complexes. Clearly, TASK-5 alone is not expressed at the plasma membrane and TASK-1 and TASK-3 assist TASK-5 channels to reach the cell surface membrane. It is noteworthy that TASK-1 and TASK-3 harbor either di-acidic forward transport signals and/or allow 14-3-3 binding, which increases surface expression. In contrast, a di-acidic motif is absent in TASK-5, and 14-3-3 did not rescue the surface expression of homomeric TASK-5 channels. However, the precise mechanism by which homodimeric TASK-5 channels are retained in intracellular compartments, presumably the ER, remains elusive, as we and others have been unable to identify a functional retention signal in TASK-5. Nevertheless, it is evident that heteromerization with another TASK family member is essential for it to be transported to the plasma membrane.

The altered pharmacology of TASK-1/TASK-5 heterodimers that we have observed for blockers as well as activators, is of uttermost relevance for future drug development. Among the activators tested, we even identified evidence for a drug that exhibits preferential activity on heteromeric TASK-1/TASK-5 channels. This may be advantageous for the treatment of PAH, given that TASK-5 in conjunction with TASK-1, is one of the most highly expressed potassium channels in pulmonary vascular smooth muscle cells (VSMCs)[16]. On the other hand, designing TASK-1 blockers that do not block heteromeric TASK-1/TASK-5 channels should be beneficial for the treatment of AFib, as this will bypass the side effect of PAH discussed for TASK-1 blockers[42,43]. In conclusion, the unique pharmacology of TASK-1/TASK-5 heterodimers must be considered for future therapy approaches and offers opportunities by specifically targeting either homomeric or heteromeric

TASK channel complexes. Our data suggest that the G95E polymorphism in *KCNK15* works synergistically in the suppression of TASK-1 and TASK-3 currents. While wild-type TASK-5 already alters the trafficking and gating of TASK-1 and TASK-3, the TASK-5[G95E] variant will cause an additional abrogation of the gating at the selectivity filter itself. The extent to which this polymorphism contributes to the suppression of TASK-1 and TASK-3 in vivo currently remains unclear, as it will depend on the transcription levels of the three different TASK channels in the respective native tissues. In addition, this polymorphism might result in a reduced number of conductive TASK-1/TASK-5 heteromeric channels at the plasma membrane in some tissue of homozygous TASK-5[G95E] 'carriers', which would result in a functional loss of those heteromeric channels with the hallmark of an enhanced drug sensitivity. Therefore, the inhibition of TASK-1 by certain blockers might result in responding or non-responding patients, depending on the targeted tissue and disease, similar to what was observed in the KOALA clinical OSA study utilizing the TASK-1 blocker BAY2586116[6] or to what is frequently observed in human medicine in general. Thus, this polymorphism needs to be carefully considered in future clinical and/or pharmacogenomics studies.

On top of that, the involvement of $K_{2P}$ channels in autoimmune diseases and cancer should not be underestimated. As with other $K^+$ channels, $K_{2P}$ and in particular TASK channels have over the last years emerged as potential targets in cancer disease[44-47]. For instance, TASK-1 and TASK-3 have been demonstrated to possess pro-oncogenic and proliferative potential in cell lines[48,49], and they are discussed as potential targets in various tumor entities[44,46,47]. Although TASK channels have been extensively studied in cancer, the role of TASK-5 in cancer remained elusive, as the channel was thought to be non-functional and/or not involved in heteromerization with other TASK channel family members. TASK-5 is dysregulated for instance in pancreatic and lung cancer, as well as in thyroid and hepatocellular carcinoma[50-53]. Thus, the knowledge that TASK-5 is engaged in heteromerization and can be characterized by its unique single-channel conductance and pharmacological properties is of major relevance to this field of research and also provides access to functionally study the role of TASK-5 in cancer.

In conclusion, our findings indicate that TASK-5 is engaged in the formation of heteromeric channel complexes with TASK-1 and TASK-3. This discovery has an impact on proper disease understanding for all clinical studies involving any of the TASK channels. Thereby our results will have major implications for future studies of TASK channel physiology and pharmacology. In particular, the altered Gq-coupled receptor-mediated channel inhibition may be of physiological relevance in tissues where TASK-1 and TASK-5 are co-expressed. Moreover, the unique pharmacology of TASK-1/TASK-5 heterodimers is highly relevant for research and the development of novel drugs targeting atrial fibrillation, obstructive or central sleep apnea, and pulmonary arterial hypertension.

## Methods
### Animals and ethical regulations
The animal study using *Xenopus* toads was approved by the Ethics Committee of the Regierungspräsidium Giessen (protocol code V54-19c 20 15 h 02 MR 20/28 Nr.A 23/2017, approved on 12.02.2018) and complies with all relevant ethical regulations.

### Cloning and site-directed mutagenesis
Human (h)TASK-5 (AF294350.1), hTASK-1 (NM_02246.3) and hTASK-3 (AF212829.1) cDNAs were subcloned into the oocyte expression vector pSGEM, human 14-3-3ε (U28936.1) and rat 14-3-3ζ (NM_013011) in pGEM-HE and the α1 receptor in pBluescript. Mutations were introduced with the QuikChange Site-Directed Mutagenesis Kit (Agilent, catalog no 200513) following the manufacturer's instructions and confirmed by Sanger sequencing (Seqlab).

### Isolation of *Xenopus laevis* oocytes, cRNA synthesis and injection
Oocytes were obtained from anesthetized, adult and sexually mature female *Xenopus laevis* frogs and incubated in OR2 solution containing in mM: 82.5 NaCl, 2 KCl, 1 MgCl₂, 5 HEPES; pH 7.5 with NaOH, supplemented with collagenase (1.5 mg/ml) (Nordmark) to remove residual connective tissue. Subsequently, oocytes were stored in ND96 solution containing mM: 96 NaCl, 2 KCl, 1.8 CaCl₂, 1 MgCl₂, 5 HEPES; pH 7.4 with NaOH, supplemented with Na-pyruvate (275 mg/l), theophylline (90 mg/l) and gentamicin (50 mg/l) at 18 °C.

TASK-1, TASK-3, TASK-5, and 14-3-3 cDNAs were linearized with NheI and cRNA was synthesized using the HiScribe T7 ARCA mRNA Kit (New England Biolabs, catalog no E2065S). cDNA of the α₁ receptor was linearized with BamHI and cRNA synthesis was done with the mMESSAGE mMACHINE™ T3 Kit (Invitrogen, catalog no AM1348). Quality was tested using agarose gel electrophoresis and cRNAs were quantified by a spectrophotometer (NanoDrop, Thermo Fisher Scientific). Stage IV and V oocytes were each injected with 50 nl of cRNA.

### Two-electrode voltage-clamp recordings
All two-electrode voltage-clamp recordings were performed at room temperature (20–22 °C) with an Axon Axoclamp 900A Microelectrode Amplifier (Molecular Devices) and a Digidata 1440 Series (Axon Instruments) as an analog/digital converter or with a TurboTEC 10CD (npi) amplifier and a Digidata 1200 Series (Axon Instruments). Micropipettes were made from borosilicate glass capillaries (GB 150TF-8P, Science Products) and pulled with a DMZ-Universal Puller (Zeitz). Recording pipettes had a resistance of 0.5–1.0 MΩ when filled with 3 M KCl solution. ND96 (pH 7.5) was used as a recording solution. Block/activation was analyzed with a voltage-step protocol from a holding potential of −80 mV. A first test pulse to 0 mV of 1 s duration was followed by a repolarizing step to −80 mV for 1 s directly followed by another 1 s test pulse to +40 mV. The sweep time interval was 10 s. Current amplitudes were analyzed at +40 mV after applying a ramp protocol. From a holding potential of −80 mV voltage was ramped from −120 mV to +45 mV within 3.5 s. Hill plots were used to calculate the half-maximal inhibitory concentration (IC₅₀) and the pH at half-maximal inhibition (pH₅₀) was calculated using the Boltzmann equation. Data were acquired with Clampex 10 (Molecular Devices) and analyzed with Clampfit 10 (Molecular Devices) and Origin 2016 (OriginLab Corp.).

Temperature sensitivity recordings were performed by using the TC-10/20 temperature controller (npi, Tamm, Germany). Usage of the HPT-2A heated perfusion tube (ALA Scientific Instruments, New York, United States) and the TS-200 miniature thermistor probe (npi, Tamm, Germany) in the recording chamber allowed for the precise heating of the bath solution to the desired temperature value. Temperature sensitivity was recorded using a ramp protocol, starting from a holding potential of −80 mV. Voltage was ramped from −120 mV to +45 mV within 3.5 s. The sweep time interval was 10 s. Temperature-dependent current changes were continuously recorded in a range between 15 °C and 35 °C and analyzed at the end of the voltage ramp (+45 mV). Differences in temperature sensitivity between homodimers and heterodimers were quantified and compared by calculating the temperature coefficient ($Q_{10}$ value) for different temperature ranges:

$$Q_{10} = \frac{I_2}{I_1}^{\frac{10\,^\circ C}{T_2 - T_1}} \tag{1}$$

### Inside-out single-channel patch-clamp recordings
Single-channel patch-clamp recordings in the inside-out configuration of manually devitellinized *Xenopus leavis* oocytes were performed at room temperature 24–48 h after cRNA injection (5 pg TASK-3 and/or 125 pg TASK-5 per oocyte). Patch-pipettes were pulled from

borosilicate glass capillaries (GB 150TF-8P, Science Products) using a DMZ-Universal Puller (Zeitz) and had resistances of 4–6 MΩ when filled with bath solution containing in mM: 140 KCl, 5 HEPES, 1 EGTA; pH 7.4 adjusted with KOH/HCl. Single-channel currents were amplified with an Axopatch 200B amplifier (Axon Instruments) and recorded with pClamp10 software (Axon Instruments) at a sampling rate of 15 kHz and the analog filter frequency set to 5 kHz using a Digidata 1550B A/D converter (Axon Instruments). Single-channel analysis was performed and the data was subsequently filtered with a 3 dB 8-pole Bessel filter at 2 kHz for illustrations using Clampfit10 (Axon Instruments). A mono-exponential fit was employed to calculate the open times and a bi-exponential fit served to calculate the closed times. A linear fit was employed to calculate the single-channel conductance.

### Quantification of surface expression

An extracellular hemagglutinin (HA) tag followed and preceded by a PGG sequence was introduced in human TASK-1, TASK-3, and TASK-5 at amino acid position 214. Surface expression of HA-tagged channel constructs was analyzed in *Xenopus laevis* oocytes 48 h after cRNA injection. To block unspecific binding of antibodies, oocytes were incubated in ND96 solution supplemented with 1% (w/v) bovine serum albumin (BSA) at 4 °C for 30 min. Subsequently, oocytes were incubated for 1 h at 4 °C with rat anti-HA antibodies (dilution 1:100, clone 3F10, catalog no #11867423001, lot #34502100, Roche), washed at 4 °C with 1% BSA/ND96 for 30 min and incubated for 30 min at 4 °C with peroxidase-conjugated secondary anti-rat antibodies (dilution 1:500, catalog no #112-036-062, lot #78598, Dianova). After washing for 1 h at 4 °C in 1% BSA/ND96 and for 15 min in ND96 solution, chemilumines-cence of single oocytes was measured as relative light units (RLUs) using SuperSignal Elisa Femto solution (ThermoFisher, 37074) and a luminometer (Promega). Non-injected oocytes served as a control.

### Confocal microscopy

For confocal microscopy, HEK293T cells were transfected in 6 cm dishes with 0.5 µg pEGFP-C1/hTASK-5 and 0.5 µg pcDNA3. On the next day, cells were seeded on poly-lysine-coated coverslips. Cells were imaged 2 days post-transfection using the 488 nm line of an argon laser on a Leica SP5 confocal microscopy equipped with a 63×/1.4 oil immersion lens.

### Drugs

Drugs were resolved in DMSO or $H_2O$ (ruthenium red) and added to the ND96 recording solution directly before recordings. Final DMSO content did not exceed 0.1%. Methoxamine hydrochloride and PD118057 were obtained from SIGMA, BAY1000493, and BAY2586116 from Bayer AG, A293, and A1899 from Sanofi GmbH, and ruthenium red, ML67-33, NS11021, BL-1249 and NS3623 from Tocris.

### Statistics & reproducibility

All values are expressed as means ± s.e.m. For all oocyte experiments, $N \geq 3$ different batches (*Xenopus laevis* toads) were used. The number (*n*) of distinct samples (oocytes with independent cRNA injections) is presented in the figures (biological replicates). The sample size was not predetermined, and the number of required experiments was estimated based on previous experiments and literature within this field. Additionally, no exclusion criteria were established, nor were any data excluded from the subsequent analysis. With regard to the experiments, no randomization or blinding was performed. The normality of the dataset was evaluated using the Shapiro–Wilk test, after which the equality of variance was assessed through the implementation of either the parametric Levene's test or the non-parametric Brown–Forsythe's test. The significance of the results was determined through the application of either the two-sided Student's *t*-test or the Mann–Whitney U-test, depending on whether the data exhibited a normal or non-normal distribution, respectively. In the event that the

variances of the data set were found to be significantly different, the statistical significance of the data set was probed with two-sided Welch's *t*-test. Conversely, for data that were not normally distributed, the Mood's median test was employed. All data are presented as mean ± s.e.m. The statistical analysis was performed using Microsoft Excel 2013 and OriginPro 2016. The respective graphs illustrate the number of biological replicates (*n*) from distinct samples. *p*-values are indicated within the figures.

### Reporting summary

Further information on research design is available in the Nature Portfolio Reporting Summary linked to this article.

## Data availability

The authors declare that the data supporting the findings of this study are available within the paper and its supplementary information files. Source data are provided with this paper.

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

## Acknowledgements

This work was supported by Deutsche Forschungsgemeinschaft (DFG) grant DE1482-9/1 to N.D. We thank Simone Preis for expert technical assistance and Michael Hahn for providing BAY compounds. The work of S.K. was supported by the Dr. Hans Messer and Dr. Rolf Schwiete Foundation.

## Author contributions

Experiments were designed by N.D. and S.R. Two-electrode voltage-clamp recordings were performed by S.R. and F.S. cDNA constructs were done by S.R.. S.R. and F.S. performed chemiluminescence assays to quantify surface expression. K.V. performed single-channel record-ings and analysis. S.R. and A.K.K. prepared figures. S.R. and N.D. drafted the manuscript. T.M. provided compounds and helped discussing and writing the manuscript. S.Sch. performed temperature sensitivity mea-surements. C.K. performed confocal imaging experiments. A.K.K. was involved in discussions. S.K. helped discussing and writing the manu-script. M.S. analyzed expression patterns and helped discussing and writing the manuscript. Each author approved the submitted version of the manuscript.

## Funding

## Competing interests

N.D. and S.R. have applied for a patent on an assay to identify heteromer-versus homomer-specific blockers to develop drugs with beneficial pharmacological properties, depending on the target disease. The remaining authors declare no competing interests.
