## [Peer Review File · Nature Communications]

Potassium channel TASK-5 forms functional heterodimers with TASK-1 and TASK-3 to break its silenceREVIEWER COMMENTS

Reviewer #1 (Remarks to the Author):

K2P15.1 (TASK-5) is notable as a K2P channel 'non-functional' subunit. In this manuscript Rinné et al. provide convincing evidence that TASK-5 can form heterodimers with the two functional members of this subfamily, K2P3.1 (TASK-1) and K2P9.1 (TASK-3). Notably, both TASK-1 and TASK-3 form functional homodimers and functional heterodimers with each other. The authors provide surface labeling evidence that extracellularly tagged TASK-5 subunits can be found on the surface of cells when TASK-1 or TASK-3 subunits are expressed, and that the presence of TASK-5 also lowers activity. The most compelling evidence for heterodimer formation are the pharmacology experiments in which the authors show that TASK-5 has effects on both TASK-1 and TASK-3, and importantly, that TASK-5 mutants, predicted to be in a drug binding site, affect drug responses. Overall, the work is interesting and important. There are a number of issues that should be addressed to improve comprehensibility and clarity.

1) One important issue that is not entirely clear is whether the authors think that the TASK-5 heterodimers are non-functional, or just have a very low activity. The data seem to support the later interpretation, but this point is not stated clearly in the manuscript, and some of the data appear to support both possibilities. The main result that has me puzzled is the lack of change in the pH responses (Fig. 3a and 3f). This result seems to be at odds with the clear changes that happen for GPCR modulation and drug binding (Fig. 4). It may be that there are no differences in key amino acids required for the pH responses. But if so, that should be stated.

2) Adding other labels to Figure 1 would help. For example, labeling the HA sphere (which is color coded with the subunit) with the tagged subunit name (ex. HA-T1 or HA-T3) would help the reader understand the figure. Also, making a plot showing the correlation between current and surface expression would be useful. They seem to be appropriately anti-correlated when TASK-5 is present (ex. 5 ng T1/T5 gives low currents in 1a and high RLUs in 1g). I think the authors are missing a chance to make a better case with their data.

3) The Ruthenium Red effect is not surprising. TASK-5 has an 'E' at the Keystone inhibitor site (just like TASK-3). This position is known to be important for Ruthenium Red binding. Thus, a TASK-3/TASK-5 heterodimer is expected to have a Ruthenium Red competent site. By contrast, TASK-1 has a 'K' at this position. The heterodimer mixture (K/E) is present in TASK-1/TASK-3 (not RR sensitive) (PMID: 1260677) and would also be present in TASK-1/TASK-5 heterodimers. Thus, the lack of an effect in the TASK-1/TASK-5 heterodimer is expected. Since a lot is known about the details of this site (PMID: 32059793) and how to interpret the data, the authors should give a clearer explanation, even if the result is as expected.

4) The supplemental results is distracting. Even though much of this section are negative results, it is strange to present them in this matter. The authors should figure out how to work the main points into the text.

5) The authors should reference the original characterization of ML67-33 (PMID: 23738709).

Reviewer #2 (Remarks to the Author):

In this manuscript, Rinné and colleagues provide the first evidence for the physiological role of TASK5, a K2P isoform previously believed to be non-functional. The authors confirm prior work showing that TASK5 homomeric channels are retained intracellularly and do not produce meaningful currents at the plasma membrane. Interestingly, they find that TASK1/TASK5 or TASK3/TASK5 heteromeric channels exhibit distinct electrophysiological and pharmacologic behaviors, as compared to TASK1 or TASK3 homomers. Overall, this electrophysiological driven study is well conceived, rigorously controlled, and motivated by a very appropriate and thoughtful interest in the future medical applications of TASK pharmacology. I enjoyed reading the manuscript and I believe this study will motivate others in the field to more closely consider the role of TASK5 in a variety of physiological contexts.

I have a few minor comments or questions that may help improve the manuscript, though these issues do not significantly detract from the overall quality of the study.

1) Do the HA tagged versions of the TASK1 or TASK3 channels function identically to wildtype? If controls for this are performed in another study, this can be cited. If not, appropriate basic controls comparing TASK1 and TASK3 gating with/without the HA tag could be useful.

2) For the single channel recordings, how did the authors determine that there are two channels in a patch (T3+T3/T5) vs a single channel that exhibits a fully open and a sub-conductance state? I ask this as it appears that even in the T3 alone recordings, there are brief openings to the lower 5 pA level while in the T3/T5 recordings, there are brief excursions to the higher conductance level. For the recordings categorized as T3+T3/T5, do the authors ever see evidence of both T3 and T3/T5 channels open at the same time? If so, this would be helpful to show in a supplemental figure, to support the idea that there are in fact two channels present in these mixed recordings. Otherwise, it seems possible that they could be observing two different modes (T3+T3/T5 and T3/T5) for what is in fact a single population of T3/T5 heterodimer.

3) In figures 3 and 4, the authors refer to specific residues known to play structural roles in TASK gating. It would be useful to include a figure showing the positions of the R7, L122 and R131 mutations in the structure of TASK1 (or TASK3), to clarify the role these positions play in forming the X-gate, latch, etc.

Reviewer #1:

K_{2P}15.1 (TASK-5) is notable as a K_{2P} channel 'non-functional' subunit. In this manuscript Rinné *et al.* provide convincing evidence that TASK-5 can form heterodimers with the two functional members of this subfamily, K_{2P}3.1 (TASK-1) and K_{2P}9.1 (TASK-3). Notably, both TASK-1 and TASK-3 form functional homodimers and functional heterodimers with each other. The authors provide surface labeling evidence that extracellularly tagged TASK-5 subunits can be found on the surface of cells when TASK-1 or TASK-3 subunits are expressed, and that the presence of TASK-5 also lowers activity. The most compelling evidence for heterodimer formation are the pharmacology experiments in which the authors show that TASK-5 has effects on both TASK-1 and TASK-3, and importantly, that TASK-5 mutants, predicted to be in a drug binding site, affect drug responses. Overall, the work is interesting and important. There are a number of issues that should be addressed to improve comprehensibility and clarity.

Thank you very much for reviewing our manuscript and for your very positive comments. We are grateful for your suggestions to improve the comprehensibility and clarity of our manuscript.

1) One important issue that is not entirely clear is whether the authors think that the TASK-5 heterodimers are non-functional, or just have a very low activity. The data seem to support the later interpretation, but this point is not stated clearly in the manuscript, and some of the data appear to support both possibilities. The main result that has me puzzled is the lack of change in the pH responses (Fig. 3a and 3f). This result seems to be at odds with the clear changes that happen for GPCR modulation and drug binding (Fig. 4). It may be that there are no differences in key amino acids required for the pH responses. But if so, that should be stated.

We apologize for not making this clear. In the single channel analyses section (Figure 3), we propose that the TASK-5 heterodimers are functional but have reduced single channel conductance and open probability. We now state this more clearly at the end of the single channel analysis results section.

TASK-1 and TASK-3 show differences in GPCR modulation (e.g. Putzke *et al.*, 2007, PMID: 17389142) and drug binding (e.g. Streit *et al.*, 2011, PMID: 21362619), presumably reflecting differences in their gating mechanism and/or the formation of drug binding sites at the intracellular gate. In contrast, the extracellular pH sensor is conserved in the TASK family and histidine 98 is the relevant proton sensor in TASK-1 (Lopes *et al.*, 2001, PMID: 11358956) and TASK-3 (Rajan *et al.*, 2000, PMID: 10747866; Kim *et al.*, 2000, PMID: 10734076). Since the residue histidine 98 is also conserved in TASK-5, we think it is not too surprising that TASK-5 does not alter the pH sensitivity of TASK-1 or TASK-3 when it is part of the heteromeric channel complex. As requested, we now state that there are no differences in the key amino acid required for pH responses in the TASK family.

2) Adding other labels to Figure 1 would help. For example, labeling the HA sphere (which is color coded with the subunit) with the tagged subunit name (ex. HA-T1 or HA-T3) would help the reader understand the figure. Also, making a plot showing the correlation between current and surface expression would be useful. They seem to be appropriately anti-correlated when TASK-5 is present (ex. 5 ng T1/T5 gives low currents in 1a and high RLUs in 1g). I think the authors are missing a chance to make a better case with their data.

Thank you for your excellent suggestions. We were relying solely on the colour coding, which was certainly not sufficient. We now also associate the HA sphere with the subunit name (e.g. T1-HA or T3-HA). This has been corrected for the former Figure 1 (now Figure 2), but also for the former Figure 3 (now Figure 4) and the new Figure 1.

In the revised manuscript, we have included a plot of relative current versus relative surface expression that includes the data from Figure 1a (T1 current) and Figure 1g (T5 surface expression), as suggested.

In addition, data of Figure 1e (T1 surface expression) was considered. In the correlation plots you can see both, the correlation that decreased amplitudes correlate with decreased currents, but also the anti-correlated surface expression of TASK-1 and TASK-5, i.e. increasing TASK-5 decreases surface expression of TASK-1, while increasing TASK-1 increases surface expression of TASK-5. These correlations are now also mentioned in the Results section and the new Supplementary Figure 4 to make a better case for our data, as you suggested.

The data have been "only" included in the Supplementary Figures section because there are caveats to this type of analysis. For example, the relationship between cRNA amount and current amplitude is not always linear, and the ELISA-based surface expression data is not linear with the amount of cRNA injected, as it depends on the accessibility of the antibody and the presumably also non-linear turnover rate for the enzymatic peroxidase reaction. However, the direction of the effects can be described semi-quantitatively with these assays and we are grateful for your suggestion as we can now make a much better case for these clear macroscopic correlations.

3) The Ruthenium Red effect is not surprising. TASK-5 has an 'E' at the Keystone inhibitor site (just like TASK-3). This position is known to be important for Ruthenium Red binding. Thus, a TASK-3/TASK-5 heterodimer is expected to have a Ruthenium Red competent site. By contrast, TASK-1 has a 'K' at this position. The heterodimer mixture (K/E) is present in TASK-1/TASK-3 (not RR sensitive) (PMID: 1260677) and would also be present in TASK-1/TASK-5 heterodimers. Thus, the lack of an effect in the TASK-1/TASK-5 heterodimer is expected. Since a lot is known about the details of this site (PMID: 32059793) and how to interpret the data, the authors should give a clearer explanation, even if the result is as expected.

Thank you for this suggestion. We have inserted the interpretation of the data and citation of the relevant literature as requested. If you agree, we have included some of your wording here. Thank you very much.

4) The supplemental results is distracting. Even though much of this section are negative results, it is strange to present them in this matter. The authors should figure out how to work the main points into the text.

Thank you for that point. We agree that the data are very valuable and that it is distracting to place them in the Supplementary Results. As also requested by the editorial team, we have included the entire former section with its Figure in the main text and Figure 1. Only the sequence alignment, which was part of the Supplementary Information Results section, is now placed as Supplementary Figure 1.

5) The authors should reference the original characterization of ML67-33 (PMID: 23738709).

We apologize for not properly citing the work with ML67-33 which we did in the revised manuscript.

Reviewer #2:

In this manuscript, Rinné and colleagues provide the first evidence for the physiological role of TASK5, a K_{2P} isoform previously believed to be non-functional. The authors confirm prior work showing that TASK5 homomeric channels are retained intracellularly and do not produce meaningful currents at the plasma membrane. Interestingly, they find that TASK1/TASK5 or TASK3/TASK5 heteromeric channels exhibit distinct electrophysiological and pharmacologic behaviors, as compared to TASK1 or TASK3 homomers. Overall, this electrophysiological driven study is well conceived, rigorously controlled, and motivated by a very appropriate and thoughtful interest in the future medical applications of TASK pharmacology. I enjoyed reading the manuscript and I believe this study will motivate others in the field to more closely consider the role of TASK5 in a variety of physiological contexts.

I have a few minor comments or questions that may help improve the manuscript, though these issues do not significantly detract from the overall quality of the study.

Thank you very much for reviewing our manuscript and for your very positive comments. We appreciate your comments and questions to improve the manuscript.

1) Do the HA tagged versions of the TASK1 or TASK3 channels function identically to wildtype? If controls for this are performed in another study, this can be cited. If not, appropriate basic controls comparing TASK1 and TASK3 gating with/without the HA tag could be useful.

Thank you for pointing out that we did not provide this information in our manuscript. We apologize for not providing the citation that the TASK-1 and TASK-3 constructs with the respective extracellular HA-epitopes are non-conductive. We have included the respective citation of Zuzarte *et al.*, 2006 in the revised manuscript (PMID: 19139046).

Unfortunately, non-functionality is very common for extracellular HA-tagged potassium channels as the extracellular linkers are very limited in size and there are only a few opportunities to introduce an HA-epitope in a way that the epitope is also accessible to the HA-antibodies. We have found a position to introduce the HA-epitope in a way that is recognized by the anti-HA antibody and further improved the accessibility of the antibody by subsequently introducing flexible proline-glycine-glycine (PGG) spacers before and after the HA epitope. It is likely that the epitope interferes with the potassium exit pathway of the extracellular ion portal (EIP), which leads to non-conducting channels. Although it would certainly be desirable to have such a functional construct with an HA-tag in which potassium conduction is not affected by the HA epitope, the purpose of the constructs was to study changes in the surface expression of TASK-1 or TASK-3, for which they served perfectly in our eyes.

2) For the single channel recordings, how did the authors determine that there are two channels in a patch (T3+T3/T5) vs a single channel that exhibits a fully open and a sub-conductance state? I ask this as it appears that even in the T3 alone recordings, there are brief openings to the lower 5 pA level while in the T3/T5 recordings, there are brief excursions to the higher conductance level. For the recordings categorized as T3+T3/T5, do the authors ever see evidence of both T3 and T3/T5 channels open at the same time? If so, this would be helpful to show in a supplemental figure, to support the idea that there are in fact two channels present in these mixed recordings. Otherwise, it seems possible that they could be observing two different modes (T3+T3/T5 and T3/T5) for what is in fact a single population of T3/T5 heterodimer.

Thank you for this excellent question and the idea that heteromeric T3/T5 channels could have two conductance states. Please note that the apparently smaller short events you noticed in the T3 recordings are still larger than the 5 pA level. However, it is difficult and not accurate enough to evaluate the current amplitudes from a single opening event, as you need to analyze the average amplitude not the short noisy peaks of a single event, which can sometimes be a bit smaller than the

average current amplitude (here the case for the first short T3 single channel event) or larger (here the beginning of the T3/T5 burst). However, we agree that the average peak (plateau) of the first T3 single-channel event illustrated in the zoom-in is slightly smaller than the auxiliary line for the large amplitude, but larger than the average amplitude of the T3/T5 burst, which is on the auxiliary line for the small amplitude. Nevertheless, when the average amplitudes of T3 events are analyzed, as we have done in panel b, there is no evidence for an additional conductance of about 5 pA, which is consistent with the absence of such a subconductance state for T3 in the literature. Regarding the idea that heteromeric T3/T5 channels could have two conductance states: In fact, we can be sure that these patches, which we think contain T3 plus T3/T5, did not have only one channel with a subconductance state, because we sometimes observed cumulative opening events of two channels, for which we now show an example in the revised Supplementary Figure 5, as requested.

3) In figures 3 and 4, the authors refer to specific residues known to play structural roles in TASK gating. It would be useful to include a figure showing the positions of the R7, L122 and R131 mutations in the structure of TASK1 (or TASK3), to clarify the role these positions play in forming the X-gate, latch, etc.

Thank you for this excellent suggestion. For the revised manuscript, we have illustrated the positions of R7 and R131 in the respective X-gate and latch structures of TASK-1 and the position of the key residue for drug binding L122 in the central cavity. These illustrations have been included in the new Supplementary Figure 2.

REVIEWERS' COMMENTS

Reviewer #1 (Remarks to the Author):

The authors have address the key points raised in the prior review. There are two stylistic points that should be addressed to improve the revision.

The abstract provides very little information for the casual reader about the key findings in the manuscript. It would seem important to have a one or two sentence summary of the main findings.

The revised results (p. 4) is a rough read. The authors jump straight to Fig. 1d, with no explanation of the contents of panels 1a-c. Also, this initial section hits the reader as a long list of things that do not affect the 'non-function'. I understand that it is important to lead with this information, but I think it could be packaged a bit better.

Reviewer #2 (Remarks to the Author):

The authors have adequately addressed my concerns and I recommend for publication in Nature Communications

Reviewer #1:

The authors have address the key points raised in the prior review. There are two stylistic points that should be addressed to improve the revision.

Thank you again for reviewing and helping to improve our manuscript.

The abstract provides very little information for the casual reader about the key findings in the manuscript. It would seem important to have a one or two sentence summary of the main findings.

Thank you for your suggestion. We have taken your suggestion on board, while staying within the 150 word limit. Please note that we have also expanded the Introduction to have some more background and rationale for the work, together with a brief summary of the major results and conclusions of the current work, as requested by the Editorial team.

The revised results (p. 4) is a rough read. The authors jump straight to Fig. 1d, with no explanation of the contents of panels 1a-c. Also, this initial section hits the reader as a long list of things that do not affect the 'non-function'. I understand that it is important to lead with this information, but I think it could be packaged a bit better.

Thank you for your suggestion and for pointing out that we did not refer to the first panels correctly. We are now explaining the panels in Figures 1a-c, while at the same time repackaging the section so that despite of including additional information the section is now one and a half double-spaced pages instead of two.

Reviewer #2:

Thank you again for reviewing and helping to improve our manuscript.